# Convolutional Forecasting of Particulate Matter: Toward a Data-Driven Generalized Model

**Luca Ferrari** [1,2,*] and **Giorgio Guariso** [2]

1 Department of Environmental Systems Science, Institute of Terrestrial Ecosystems, ETH Zurich, 8092 Zurich, Switzerland

2 Department of Electronics, Information, and Bioengineering, Politecnico di Milano, 20133 Milan, Italy; giorgio.guariso@polimi.it

* Correspondence: luca.ferrari@usys.ethz.ch

**Abstract:** Air pollution poses a significant threat to human health and ecosystems. Forecasting the concentration of key pollutants like particulate matter can help support air quality planning and prevention measures. Deep learning methods are becoming increasingly popular for predicting air pollution and particulate matter concentration. Architectures like Convolutional Neural Networks can effectively account for the geographical features of the study domain. This work tests a Feed-Forward, a Long Short-Term Memory (LSTM), and a Convolutional Neural Network (CNN) on a polluted geographical domain in northern Italy. The best convolutional architecture was then implemented in two other quite different regions. The results show that the same CNN architecture provides remarkably accurate forecasts in all applications and that a network trained on $PM_{10}$ data can accurately forecast $PM_{2.5}$ concentrations up to 10 days ahead. These results suggest that the proposed CNN has high generalization capabilities and can thus be reliably used as a forecasting model for different areas.

**Keywords:** air quality; particulate matter; deep learning; multi-step forecasting; generalization

## 1. Introduction

Air pollution is a relevant environmental issue with global consequences [1]. Its impact spans across several areas: increasing mortality in forests [2], declining agricultural productivity [3], and impacting and reducing biodiversity overall [4].

Amongst the most relevant effects, air pollution is known to represent the most significant environmental threat to human health on a global scale [5]. Global ambient particulate matter (PM) concentration has increased over the past few years [6]. Such increases have been associated with an increase in all-cause mortality [7], cardiovascular disease [8,9], and cancer [9,10]. Overall, it is estimated that diseases caused by air pollution cause up to 9 million premature deaths per year [11,12], leading to economic losses of up to 6.2% of the world's GDP [12].

Forecasting PM concentration can have relevant implications for air quality planning and the development of response measures. However, the dynamic of PM concentrations is quite complex and differs in different contexts because of the peculiarities of local emissions and meteorology. Indeed, a portion of PM can be directly produced by combustion and friction processes (primary PM, mainly constituted by larger particles with an equivalent diameter of up to 10 microns, $PM_{10}$). Another portion is formed in the atmosphere due to complex physical and chemical reactions. It is mainly constituted by finer particles (secondary PM, constituted by particles with a diameter up to 2.5 microns, $PM_{2.5}$). The gases that lead to the formation of $PM_{2.5}$ are called "precursors" and are mainly constituted by sulfur and nitrogen oxides and by volatile organic compounds (VOCs). They are emitted by various anthropic activities such as industry, domestic heating, and traffic.

Several process-based chemical transport models (CTMs) have been developed and are currently in use to simulate the evolution of primary and secondary PM concentrations (see, for instance, [13,14]). In addition to information on emissions, they also need data on the local meteorology that drives all the formation processes mentioned above.

A completely different perspective is adopted by data-driven models that forecast PM based on empirical relationships between a set of inputs (including past concentration data) and the desired output (forecast values).

Deep learning approaches are being employed with increasing popularity for this purpose [15–17]. Model structures like the Recurrent Neural Network (RNN) are among the most popular choices. Zhao et al. [18], for instance, propose an approach that allows for capturing long-term and short-term features and learning spatial and temporal properties. Convolutional Neural Networks (CNNs) are also a popular model for extracting spatial–temporal dependencies. Zhang et al. [19] use a CNN-based method that assigns weights to the inputs in the space and time dimensions to enhance the essential information. Other popular neural network structures include Graph Neural Networks (GNNs) [20] and several hybrid approaches. As Refs. [21,22] show, combining a Graph and either a Convolutional or a Long Short-Term Memory (LSTM) network can be effective when modeling the spatiotemporal variation in PM concentration and the spatial dependence between measuring stations. Other approaches also include the combination of a Convolutional and an LSTM network. The approach was first proposed for precipitation forecasting [23,24] but has also been adopted for air quality forecasting [25] since then.

Spatial variability is a typical characteristic of most environmental variables, including air pollution. Methods to take it into account vary from well-known point interpolation methods [26] to more sophisticated deep learning approaches [27–30]. In any case, the traditional method is to develop a specific neural network for each measurement site and each pollutant, as it is inherent to the data-driven concept.

In this paper, we demonstrate that this limitation can, at least in part, be overcome. While it is true that neural networks, like any other data-driven approach, need to be based on data to be reproduced, a well-designed network architecture can grasp the key features of a phenomenon. It can thus be reliably applied also in other similar circumstances. This means it somehow has the generalization capability of a process-based model and is suited for general use without searching, case by case, for the best architecture, as is typical of neural network applications. To prove this statement, we first develop a neural network for $PM_{10}$ forecasting in the domain of Lombardy in northern Italy. For this purpose, three popular architectures are compared: Feed-Forward, LSTM, and CNN. Then, the best-performing architecture is applied to two other quite different environmental domains (Lower Silesia, Poland, and Great Sydney, Australia). Finally, we also test the ability of the networks trained on $PM_{10}$ to forecast $PM_{2.5}$ concentrations in the same regions.

The excellent performances shown by all PM predictors up to 10 days ahead show that, with a sufficiently long and rich training period, the proposed architecture can embed the necessary knowledge to become a general PM forecasting model that can be used in all cases.

The rest of this paper is organized as follows: Section 2 presents the data and the regions of this study, and describes the methods and the structures of the neural networks we developed. Section 3 shows and comments on the numerical results, while Section 4 concludes this paper by summarizing the main findings.

## 2. Materials and Methods

### 2.1. Data

Time series of average daily concentration data for $PM_{10}$ and $PM_{2.5}$ were collected for 2015–2021 in three different contexts. In all areas, $PM_{10}$ daily data between 2015 and 2019 were used to train the networks, 2020 data for validation (validation dataset), and 2021 data for testing the generalization ability of the networks (test dataset).

Some data were missing in every context. We reconstructed the missing data using two methods: For single missing days, we estimated the value through a linear interpolation of neighboring values from the same measurement station. For larger gaps, we identified the k-nearest neighboring stations to those with missing data. Missing values were estimated using the Inverse Distance Weighting (IDW) method.

## 2.2. Study Area

### 2.2.1. Lombardy, Italy

The first domain considered is Lombardy, an almost 24,000 km$^2$ region in northern Italy with a population of about 9 M. The area is densely inhabited in the southern part and protected by the Alps to the north. The average temperature varies between 4 °C in January and 30 °C in June, with a mean yearly precipitation of 1000 mm. The orography determines a situation with low winds (average below 2 m/s) and frequent temperature inversions that often determine high-concentration episodes, particularly in winter. A total of 31 stations regularly measure PM$_{2.5}$ and 63 PM$_{10}$, with a missing value rate of around 5% (www.arpalombardia.it, accessed on 3 May 2022). They are mainly located on the southern plain, where industrial and agricultural activities are denser. In this part, the winter average daily PM$_{10}$ often exceeds the threshold of 50 μg/m$^3$, which represents the limit not to be violated more than 35 days a year, according to the current European legislation. Indeed, 52 stations out of 63 exceeded this limit in the years considered. The peak values reached 264 μg/m$^3$ for PM$_{10}$ and 182 μg/m$^3$ for PM$_{2.5}$. A map of the region with the position of the measurement stations is shown in Figure 1.

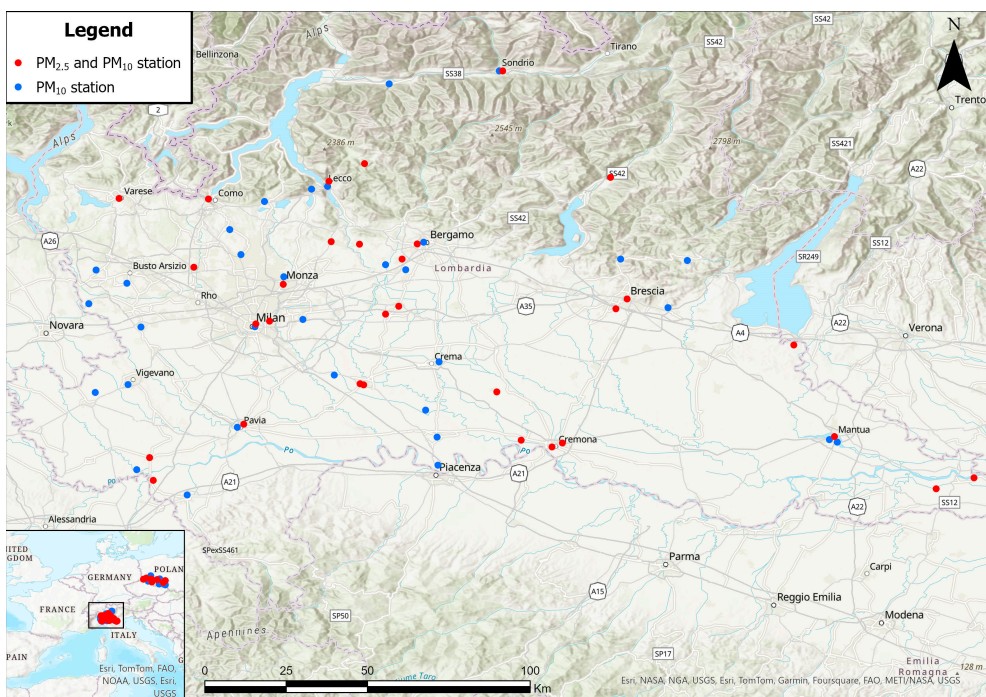

**Figure 1.** PM measuring stations in Lombardy.

To a large extent, PM$_{2.5}$ is of secondary origin and constitutes the largest portion of PM$_{10}$, a percentage of around 70% on average. According to the 2021 emission inventory (www.inemar.eu, accessed on 20 October 2023), the main sources of primary PM were domestic heating (6480 t/year for PM$_{2.5}$ and 6640 for PM$_{10}$) and traffic (2280 t/year for PM$_{2.5}$ and 3340 for PM$_{10}$). Together, these two sectors represented about 70% of the emissions of primary PM. As for the precursors of secondary particles, of the almost 95 kt/year of nitrogen oxides emitted, 45% derived from road transport, whereas the industrial use of solvents constituted one-third of the 240 kt/year of VOC emissions. Even though

the PM concentration in the region has been decreasing in the last decade [31], the region is still amongst the most polluted areas in Europe and in the world [1,6].

### 2.2.2. Lower Silesian and Opole Voivodeships, Poland

The Lower Silesian Voivodeship and Opole Voivodeship (LSOVs) (provinces) cover an area of almost 30,000 km$^2$ in southwestern Poland. With a population of about 3.9 M, these regions have a significantly lower population density than the previous site. The region is mostly flat; the Sudeten Mountains run southwest along the Polish/Czech border. The region is heavily industrialized, particularly in the southwestern area of the Lower Silesian Voivodeship. This area is part of the so-called Black Triangle, a European region across the Czech Republic, Germany, and Poland. Also, this area is known to be one of the most polluted in Europe [32]. The average temperature spans from around 3 °C in January to 25 °C in July, with a yearly precipitation of about 700 mm. The average wind speed varies between 2 and 4 m/s. A total of 7 stations measure daily $PM_{2.5}$ concentrations and 17 measure $PM_{10}$, with a rate of missing values of up to 5% (Figure 2).

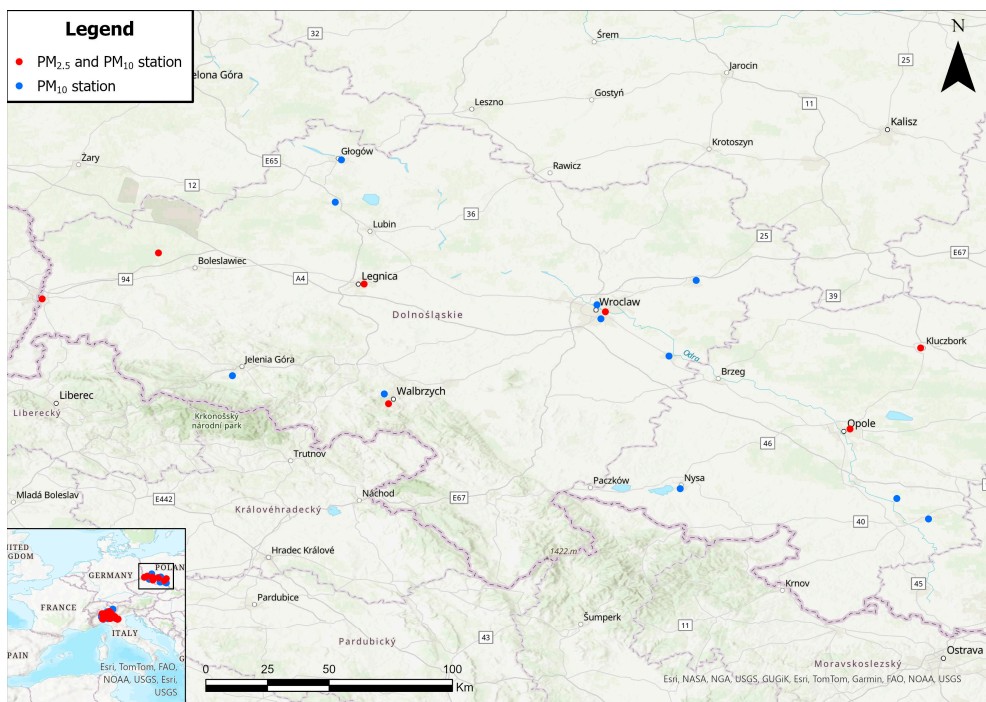

**Figure 2.** PM measuring stations in the Lower Silesian and Opole Voivodeships.

The average $PM_{10}$ concentration was around 27 μg/m$^3$ in the considered period. There is high seasonal variability, as the concentration can reach maximum values of up to 313 μg/m$^3$ in winter. Furthermore, the threshold of 50 μg/m$^3$ was violated more than 35 days per year for every year considered, with some stations measuring higher concentrations for 81 days per year on average [33]. Like in the case of Lombardy, $PM_{2.5}$ constitutes about 64% of $PM_{10}$.

However, the emission sources in Poland are different, with a high emission of sulfur dioxide mainly due to energy production(42%). Power plants were also responsible for 21% of nitrogen oxides emissions in 2020 [34]. Road transport is the main source of nitrogen oxides (35%), while its contribution to primary PM is estimated to be very low (5%). Again, primary particles are mainly produced by domestic heating (probably wood combustion), accounting for about 43%.

Long-term exposure to particulate matter in Polish cities is responsible for thousands of hospitalizations [35,36].

### 2.2.3. Greater Sydney, Australia

Sydney's metropolitan area (GSA) is the most densely populated Australian city; it is located in the New South Wales state on the east coast of the Pacific Ocean, as shown in Figure 3.

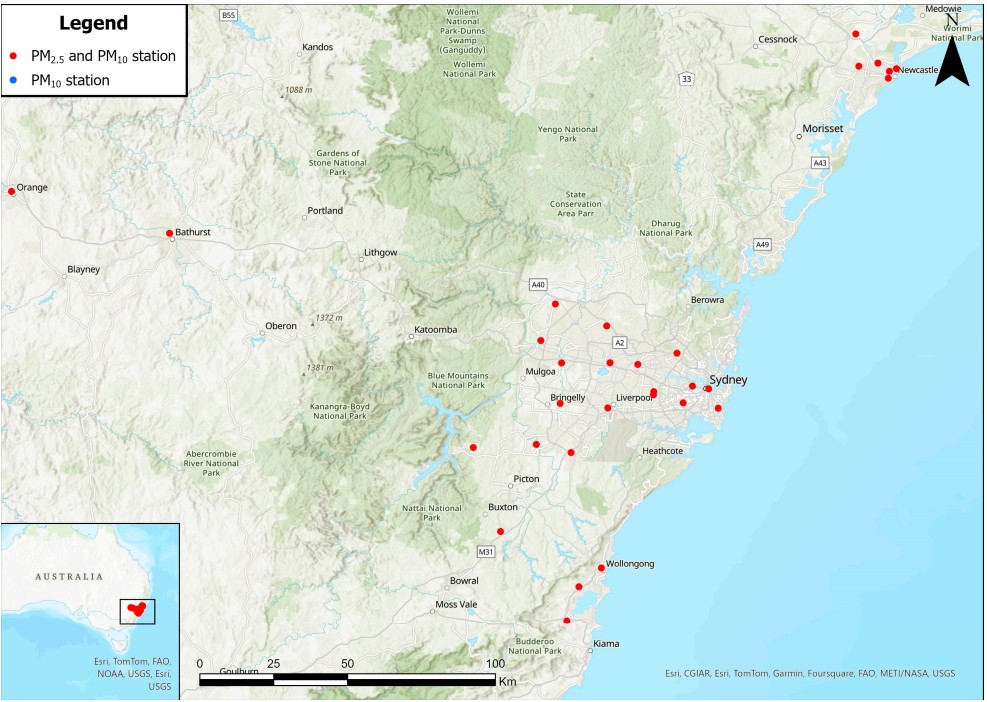

**Figure 3.** PM measuring stations in the Greater Sydney Area.

The total surface area is around 12,000 km$^2$. It occupies a flat region to the south, while the Hornsby Plateau lies to the north. It is only a few hundred meters high and thus does not influence the local wind conditions much. The average wind speed is around 5.7 m/s. The average rainfall is about 1100 mm/year, like in Lombardy. The 2021 population was slightly more than 5 M. PM$_{10}$ and PM$_{2.5}$ concentrations are measured by 30 stations, with a rate of missing data of about 4% for the stations that were not discontinued. The density of traffic and industrial emissions is lower than that in Lombardy [37], so the average PM$_{10}$ concentration was just under 20 µg/m$^3$ in the period considered. Peak values are usually reached in December–January and are around 30 µg/m$^3$. Only during one episode in January 2020 did all the stations register much higher values: the average concentration topped 155 µg/m$^3$ and remained over 100 µg/m$^3$ for several days. Again, due to the different emission and meteorological conditions, PM$_{2.5}$ is just about 22% of PM$_{10}$, with an average value of 8 µg/m$^3$.

Primary PM$_{2.5}$ emissions differ from those of the other two sites, with 40% due to domestic heating (wood burning) and 28% due to natural contributions (bushfires, marine aerosols) (www.nsw.epa.gov.au, accessed on 12 October 2023). Domestic and commercial activities are also estimated to emit 45% of VOCs' 128 kt/year, with natural emissions accounting for 25%. Road traffic contributes about 50% to the 61 kt/year of nitrogen oxides emissions.

Climate change is expected to affect air quality and human health, although there is limited knowledge related to the Australian context [38]. Events such as dust storms and bushfires, both expected to increase because of climate change, can lead to high PM$_{10}$ concentrations. This can result in a significant increase in mortality, related to both kinds of events [39]. Large dust storms like the one recorded in September 2009 can raise PM$_{10}$ concentration to thousands of µg/m$^3$ and significantly increase hospitalizations [40]. A reduction in air pollution could result in considerable health benefits for the population of

Sydney [37]. Preventive measures such as hazard-reduction burning can reduce the risk of bushfires but can cause high smoke-related PM concentrations [41]. A map of the area considered in this study with the measurement stations is shown in Figure 3.

*2.3. Methods*

Various network architectures were first tested on the denser Lombardy data, and then the most effective of them was extended to the other domains. In all instances, we used data from every station spanning the first five years (1825 values times the number of stations) for calibration. The year 2020 was used to validate the models, and the final year, i.e., 2021, was used for testing. We experimented with the classical Feed-Forward (FF), Long Short-Term Memory (LSTM), and Convolutional Neural Network (CNN) structures. As is well known, LSTMs are supposed to represent long-term trends, such as the yearly periodicity, while CNNs are best suited to catch specific configurations of values.

Several combinations of neurons, sizes of convolutional filters, and hidden units in LSTM layers were tested. We selected architectures based on their performance, measured by their ability to forecast accurately over a 10-day horizon. For the FFNN, we experimented with a two-layer architecture, exploring variations in the number of neurons: 10, 20, or 64 for the first layer and 5, 10, or 32 for the second layer. For the CNN and LSTM, we tested different filter sizes and numbers of neurons in the network. Filter sizes ranged from 5 to 20, neurons varied between 4 and 20, and the number of hidden units in the LSTM ranged from 5 to 15.

The structures finally adopted are the following: The FF Network has two hidden layers, one with 64 neurons and one with 32. As for the LSTM, we used one LSTM layer with 15 nodes, coupled with a dropout layer with a probability of 0.2, a fully connected layer, and a regression output layer. We implemented a 1-D convolutional layer for the CNN with a filter size of 20 and 20 neurons. The convolutional layer was followed by batch normalization and a ReLu layer. Lastly, we added a fully connected layer and a regression output layer to this network. We trained the networks using the Adam algorithm.

As for the input data, we used the $PM_{10}$ concentration of the current day in a selected station and the geographical coordinates (latitude and longitude) of the station, representing its geographical location as shown in Equation (1).

The output is the particulate concentration forecasted for the station for each of the following ten days. This means that the value for a specific calendar day is forecasted several times as the specific date approaches, thus with greater and greater precision.

Thus, the model has a vector output and can be written as follows:

$$\left[ \hat{y}^i_{t+1} \ \hat{y}^i_{t+2} \ \hat{y}^i_{t+3} \ \hat{y}^i_{t+4} \ \hat{y}^i_{t+5} \ \hat{y}^i_{t+6} \ \hat{y}^i_{t+7} \ \hat{y}^i_{t+8} \ \hat{y}^i_{t+9} \ \hat{y}^i_{t+10} \right] = f_a\left( y^i_t, lat^i, long^i \right) \qquad (1)$$
$$i = 1, \dots, N_a$$

where $\hat{y}^i_{t+k}$ represents the value forecast for time $t + k$ for measurement station $i$; $y^i_t$ is the current value measured at the same station, whose coordinates are $lat^i$ and $long^i$; and $f_a$ $(\cdot,\cdot,\cdot)$ is the neural model and is the same for all the $N_a$ measurement stations in each area $a$.

The networks are trained to minimize the average root-mean-square error (RMSE) of all the measurement stations in an area on all days of the current forecasting horizon. This means that

$$\mathrm{RMSE}_a = \frac{1}{N_a H_T} \sqrt{\sum_{i=1}^{N_a} \sum_{t=1}^{H_T} \left( y^i_t - \hat{y}^i_t \right)^2} \qquad (2)$$

where $H_T$ is the training horizon.

Equation (2) implies that the precision of the forecast for the following day may be somehow reduced to obtain a better performance on the nine following days.

In this way, we obtain an "area" model, i.e., a unique neural network that best performs on all the stations considered together. This differs from the classical approach, where a specific model is developed for each station.

We assessed the overall performance of each network on the test dataset using classical indicators like the normalized mean absolute error (NMAE), the normalized root-mean-squared error (NRMSE), and the coefficient of determination ($R^2$) [42]. In the test phase, the same model $f_a$ was applied to the data of each individual station as it is normally performed. The results presented later are the average of the performance metrics computed for each station.

We also evaluated a classical persistence model that we used as a comparison for neural networks. This model can be described as follows:

$$\left[ \hat{y}^i_{t+1} \ \ \hat{y}^i_{t+2} \ \ \hat{y}^i_{t+3} \ \ \hat{y}^i_{t+4} \ \ \hat{y}^i_{t+5} \ \ \hat{y}^i_{t+6} \ \ \hat{y}^i_{t+7} \ \hat{y}^i_{t+8} \ \ \hat{y}^i_{t+9} \ \ \hat{y}^i_{t+10} \right] = y_t \qquad (3)$$

Equation (3) thus assumes that the forecasted particulate concentration remains constant over the ten-day horizon.

In the test phase, all the predictors work as if they were acquiring new real-time data and issuing the forecast for each day of the year 2021.

## 3. Results and Discussion

Figure 4 presents the performance of the three neural network architectures and the persistence model over Lombardy using the test dataset. The horizontal axis represents the days of the forecasting horizon, and the vertical axis represents the three metrics to assess the models' performance: $R^2$ in Figure 4a, NRMSE in Figure 4b, and NMAE in Figure 4c. The values represent, for each model, the average performance for each forecasting step over the test dataset.

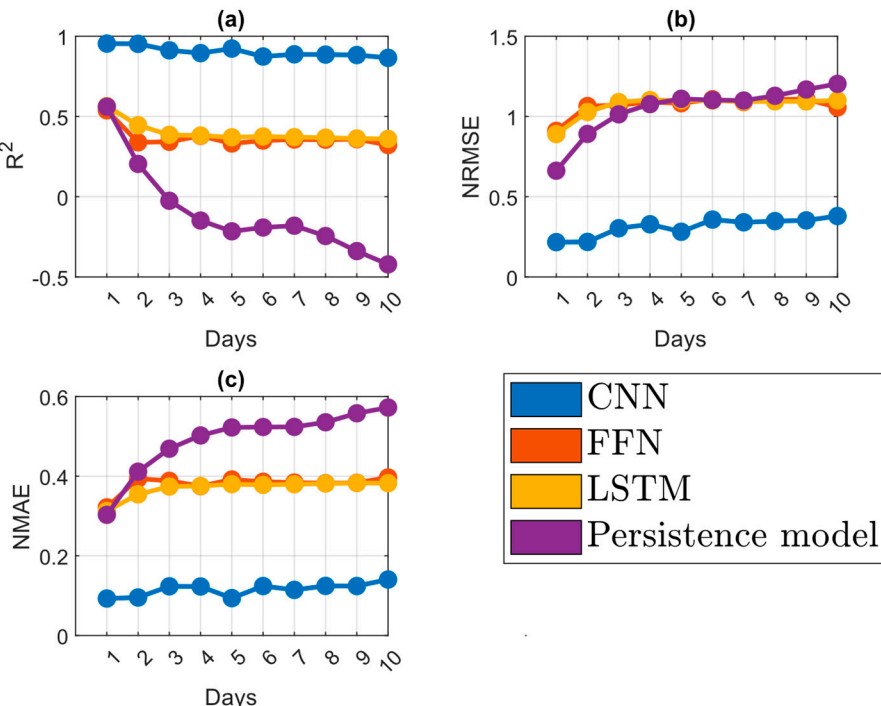

**Figure 4.** Neural networks test performances on the Lombardy domain showing the coefficient of determination $R^2$ (**a**), the NRMSE (**b**), and the NMAE (**c**).

The coefficients of determination reported in Figure 4a show that the FF and the LSTM networks produce a performance similar to the persistence model over the first days of the horizon. They reach an $R^2$ of 0.58 and 0.62, respectively, while the persistence model has an $R^2$ of 0.59 on the first day. The networks achieve better performances in the following forecasting steps: the persistence model's $R^2$ decreases and reaches negative values, while the FF and LSTM networks stabilize and show an $R^2$ of 0.35 and 0.38, respectively. The

NMAE reported in Figure 4c shows a similar pattern, with the FF and LSTM networks reaching an error of around 0.38 in the 10th time step and the persistence model reaching an error of 0.54. The NRMSE reported in Figure 4b, however, shows a different pattern, with the three models having similar error values that grow significantly in the first days and stabilize around 1.09 and 1.2 on the tenth day. The CNN has significantly better performance, corresponding to the highest $R^2$ and the lowest error. Its $R^2$ goes from 0.94 on the first day to 0.88 on the tenth day. The NRMSE and NMAE reach a maximum value of 0.35 and 0.13, respectively.

The scattergrams in Figure 5 show the high performance of the CNN over Lombardy. Here, the horizontal axis represents the actual $PM_{10}$ concentrations, and the vertical axis represents the correspondent values forecasted by the CNN over two measuring stations, in addition to the performance over the whole of Lombardy. The stations are numbers 20 and 39 out of 63. They have been randomly selected and demonstrate the different accuracies that can be obtained. Figures in the first column (Figure 5a,c,e) represent the third-day-ahead forecast, while in the second column (Figure 5b,d,f), they represent the seventh day. Each point represents the forecasted $PM_{10}$ concentration over the 2021 test dataset. The figure also reports the $R^2$ coefficient and the NRMSE. The plots show an expected slight decrease in performance when moving from the third to the seventh day and an increased but limited tendency to overestimate some low-to-medium values. Overall, the accuracy remains high and the error low, consistent with the results shown in the previous figure. Focusing on the three-day-ahead forecast, which can be considered a sufficient interval to issue alarms and undertake mitigation measures, an analysis of the errors shows that they exceed 50% of the measured value only in 2.8% of the cases. Additionally, using the traditional definition of accuracy (number of correct forecasts above or below a threshold over the total number of cases), one obtains a value of 0.97 for the legal threshold of 50 μg/m$^3$ and 0.99 for a threshold of 100 μg/m$^3$. This confirms the high performance of the model in forecasting peak pollution episodes.

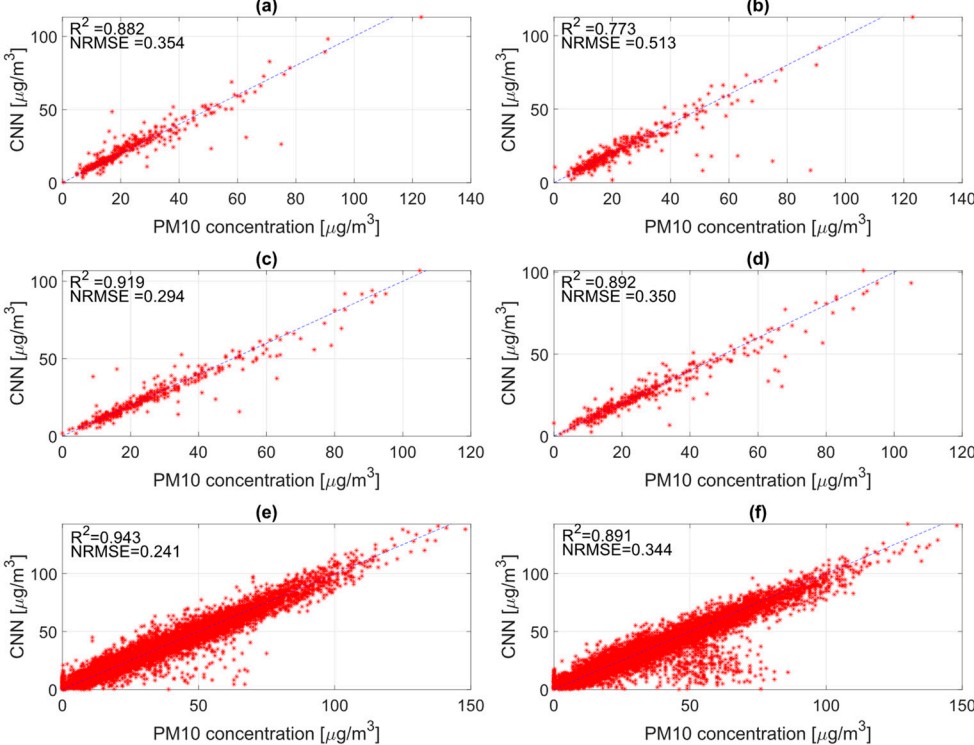

**Figure 5.** Scatterplots of forecasted values and concentration data for the CNN on Lombardy for the 3rd (**a**,**c**,**e**) and 7th (**b**,**d**,**f**) days of the forecasting horizon in randomly selected stations: station n° 20 (**a**,**b**); station n° 39 (**b**,**c**); and all the stations (**e**,**f**).

Since the FF and the LSTM networks performed with lower accuracy, the $PM_{10}$ on the other two domains was forecasted by training only the CNN. Figure 6 reports the same information as Figure 5, referring to the persistence model and the CNN over the Greater Sydney Area and Lower Silesian and Opole Voivodeships.

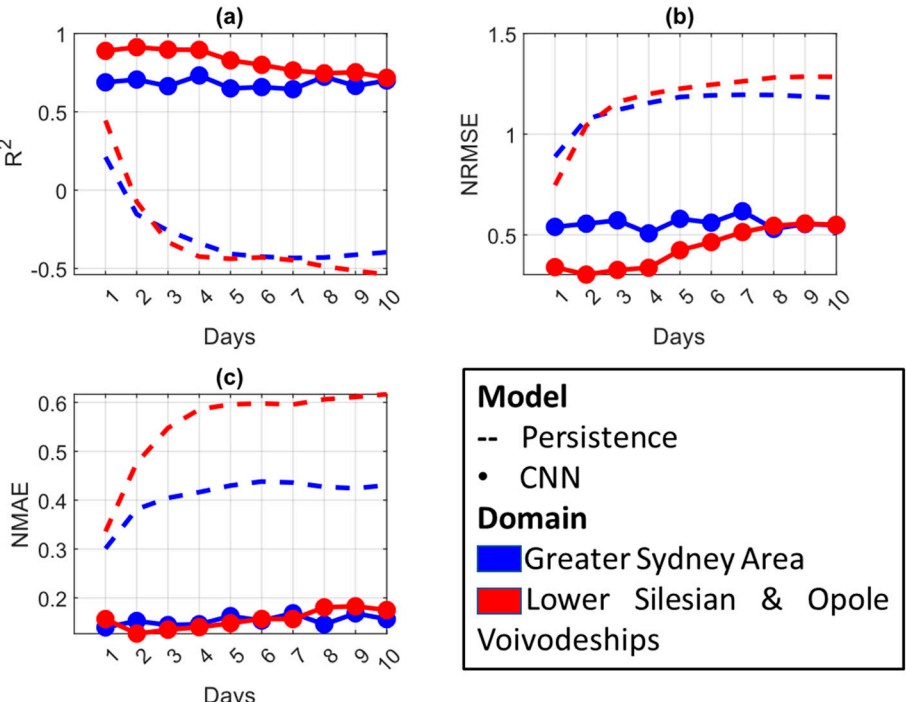

**Figure 6.** CNN test performances in Greater Sydney and the Lower Silesian and Opole Voivodeships showing the coefficient of determination $R^2$ (**a**), the NRMSE (**b**), and the NMAE (**c**).

The CNN performances are coherent with the Lombardy case, with a coefficient of determination reaching a value of 0.72 on the 10th day in the Polish case. The performance in the Greater Sydney Area is lower, with an $R^2$ between 0.64 and 0.73 over the forecasting horizon. The errors reach values of 0.55 and 0.18 on the 10th day in the Polish case and slightly lower values in the Australian case, probably because of the different composition and distribution of particulate matter and meteorological conditions.

The scattergrams in Figure 7 confirm the high performance of the CNNs. Here again, the horizontal axis represents the measured $PM_{10}$ concentrations over the test dataset, and the vertical axis represents the correspondent values forecasted by the CNNs over the domains of the Greater Sydney Area and the Lower Silesian and Opole Voivodeships. Figure 7a,c represents the third-day-ahead forecast, while Figure 7b,d represents the seventh day. The figure also reports the $R^2$ coefficient and the NRMSE. The two CNNs show high performances, corresponding to a high $R^2$ and a low error. The performance decreases when moving from the third to the seventh day, as expected. In particular, the CNN trained over the Lower Silesian and Opole Voivodeships confirms that the network can somehow underestimate low-to-mid concentration levels, as previously observed.

Since a significant portion of $PM_{10}$ is constituted by $PM_{2.5}$, the networks trained over $PM_{10}$ concentrations were tested using $PM_{2.5}$ concentrations as inputs. Figure 8 presents the performance of the neural network architectures over Lombardy, Greater Sydney, and the Lower Silesian and Opole Voivodeships. We reported the same performance metrics as in Figure 6. In this case, we used $PM_{2.5}$ concentrations over the test year of 2021 for the three regions as the test data.

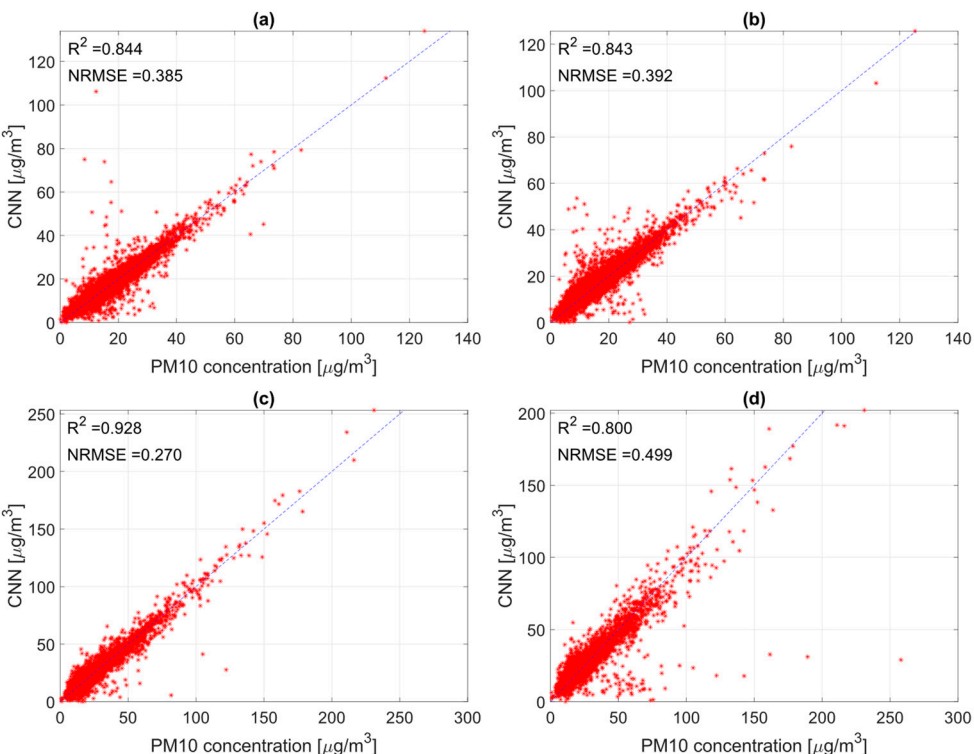

**Figure 7.** Scatterplots of forecasted values and concentration data for the CNNs on the 3rd (**a**,**c**) and 7th (**b**,**d**) days of the forecasting horizon for the domains of the Greater Sydney Area (**a**,**b**) and the Lower Silesian and Opole Voivodeships (**c**,**d**).

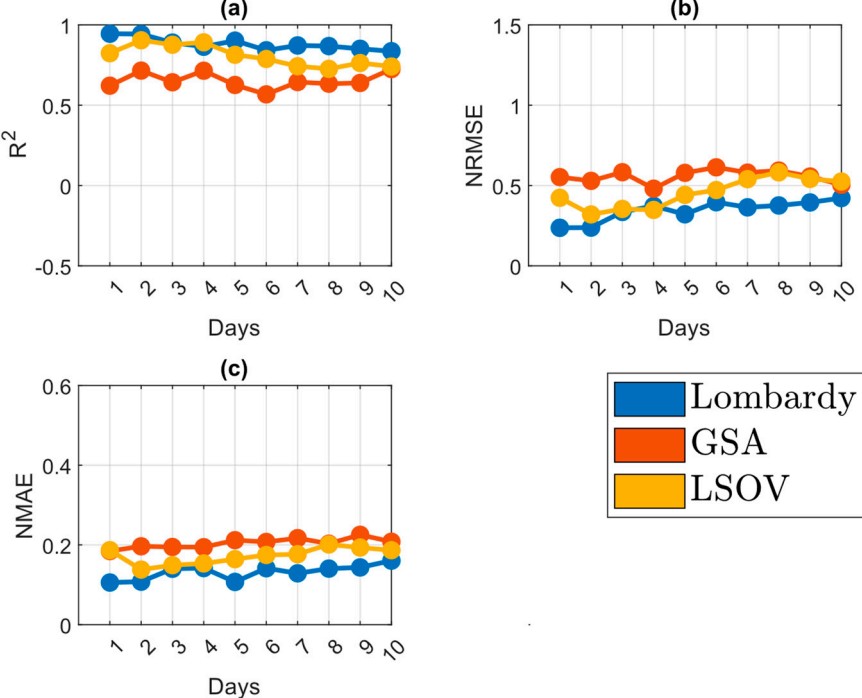

**Figure 8.** CNN performances for $PM_{2.5}$ data showing the coefficient of determination $R^2$ (**a**), the NRMSE (**b**), and the NMAE (**c**).

The CNNs show a high performance in every region. For Lombardy, the network reaches a coefficient of determination of 0.83 on the 10th day. In the GSA and LSOV areas, the networks have slightly lower performances, with a coefficient of 0.74. The $R^2$ and the

error metrics show that the CNN trained over Lombardy and the LSOV area performed better than in the Australian case, consistent with the previous results. With the highest NMAE of 0.22 and NRMSE of 0.61, the CNNs can forecast $PM_{2.5}$ concentrations with high accuracy. The performances across the three metrics are comparable with those obtained in the case of $PM_{10}$ forecasting, even if the networks were not retrained for the specific $PM_{2.5}$ case.

Lastly, we tested the ability of each CNN to forecast $PM_{10}$ concentrations on the two domains on which they had not been calibrated and validated. We aimed to assess the capability of the model to forecast $PM_{10}$ concentrations over different domains, with significant differences in geographical features and PM composition. We report the same performance metrics as in previous figures.

In the first case, we used the $PM_{10}$ concentration test data from the Great Sidney Area and the Lower Silesian and Opole Voivodeships as inputs for the CNN trained over the Lombardy domain. The results are reported in Figure 9. Again, the CNNs show high $R^2$ and low errors over the LSOV domain. In the GSA domain, we observe a decrease in performance and an increase in error values. Overall, the CNN trained over Lombardy has a performance over the Lower Silesian and Opole Voivodeships consistent with the previous ones, but it forecasts $PM_{10}$ concentrations over the GSA domain with less satisfactory accuracy.

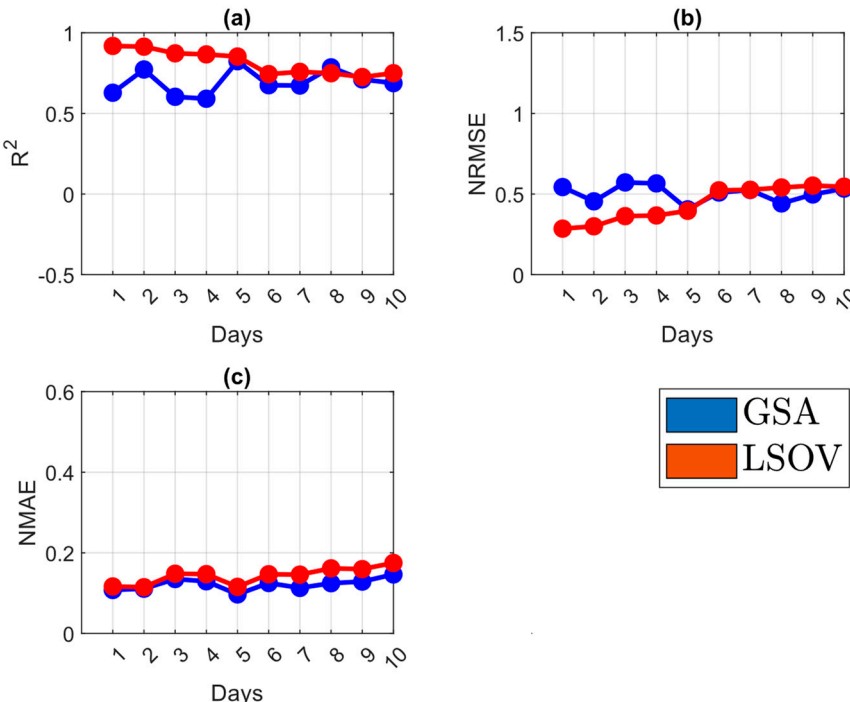

**Figure 9.** Performance of CNN trained over Lombardy, using Australian and Polish data, showing the coefficient of determination $R^2$ (**a**), the NRMSE (**b**), and the NMAE (**c**).

In the second case, we used the $PM_{10}$ concentration data from Lombardy and the Lower Silesian and Opole Voivodeships as inputs for the CNN trained over the GSA domain. The $R^2$ reported in Figure 10 shows higher performances on Lombardy than on LSOV data. The NRMSE on the 10th day reaches values of 0.44 and 0.62, respectively. The CNN trained over the GSA data offers comparable performances when applied to both its original domain and the alternative ones.

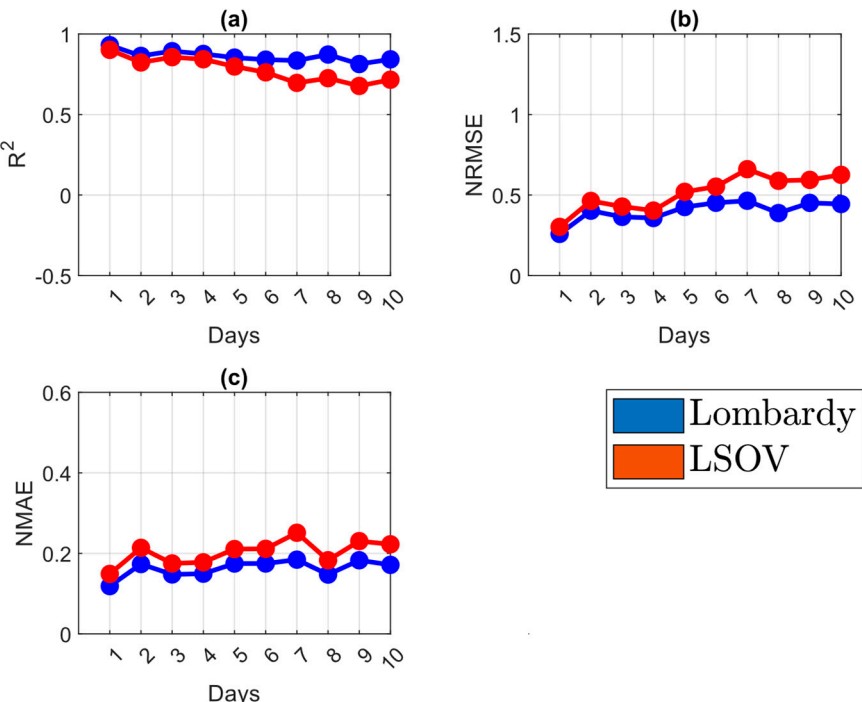

**Figure 10.** Performance of CNN trained over GSA, using data from Lombardy and LSOV, showing the coefficient of determination $R^2$ (**a**), the NRMSE (**b**), and the NMAE (**c**).

In the last case (Figure 11), we used $PM_{10}$ concentration data from Lombardy and the GSA as inputs for the CNN trained over the LSOV domain. The network performs well on the Lombardy domain, with high $R^2$ and low errors. The forecast on the GSA domain results in lower $R^2$ and higher NRMSE values, so in this case, the CNN also forecasts $PM_{10}$ concentrations with reduced accuracy on the GSA domain.

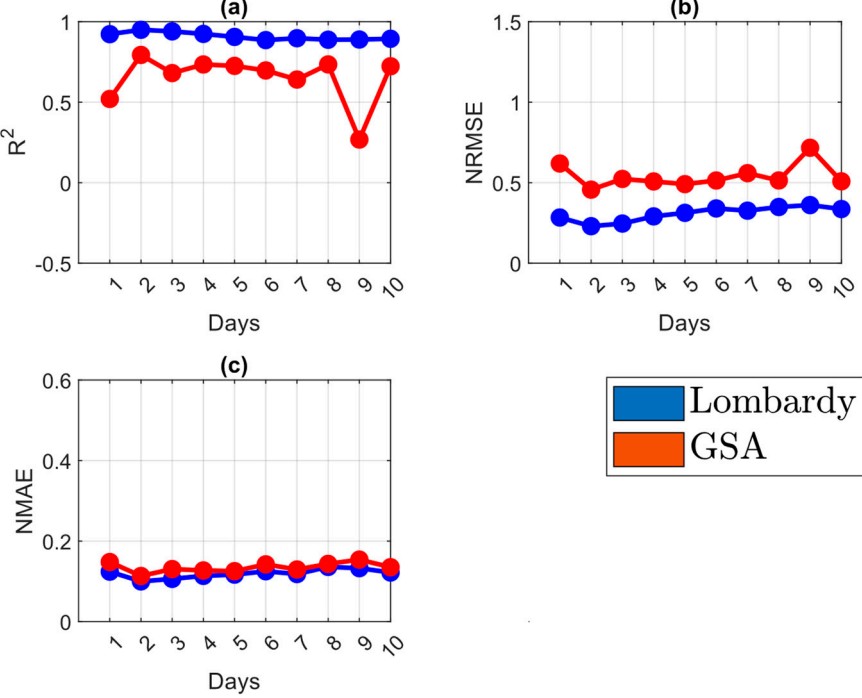

**Figure 11.** Performance of CNN trained over LSOV on Lombardy and GSA data, showing the coefficient of determination $R^2$ (**a**), the NRMSE (**b**), and the NMAE (**c**).

The different approaches vary significantly in terms of computational effort and time (see Table 1). The FFN, despite its simple structure, takes around 47 s to train. The LSTM needs a significantly longer time. However, both of these networks provided PM$_{10}$ forecasts that were significantly less accurate than those of the CNN. This network offers a significant improvement in terms of accuracy and computational effort over all three domains, as it takes between 6 and 13 s to train. These results were obtained on a computer using an AMD Ryzen 9 5900HS processor.

**Table 1.** Time required to train and run the neural networks implemented in this analysis.

| Network Type | Training Time [s] | Running Time [s] |
|---|---|---|
| FF Lombardy | 46.67 | 0.34 |
| LSTM Lombardy | 695.38 | 2.27 |
| CNN Lombardy | 13.79 | 0.03 |
| CNN Greater Sidney Area | 8.49 | 0.03 |
| CNN Lower Silesian and Opole Voivodeships | 6.02 | 0.02 |

## 4. Conclusions

Forecasting the concentration of particulate matter days ahead is a complicated yet crucial task. It can provide support to decision-makers for air quality planning and developing response measures such as warnings to reduce population exposure (particularly for sensitive classes) or decrease emitting activities such as industrial plants or traffic. Deep learning methods are becoming increasingly popular for their versatility and accuracy. This study has demonstrated that the classical approach of developing a unique forecasting model for each measurement station can be overcome by a suitable convolutional architecture. In fact, we tested three popular multi-step neural network structures and applied our findings to three different spatial domains, accounting for the spatial variability and distribution of particulate matter. In each area, a general model was developed by training the network on the data of all the measurement points and then testing it on each station separately. The high performance obtained allows the use of such models also for new stations in the area without the need for long datasets for their training.

More in detail, on the Lombardy domain, the CNN architecture provides precise results with significantly low errors, while the FF and LSTM architectures forecast PM$_{10}$ with lower accuracy. In particular, the NRMSE goes from 1.2 to 0.35 on the 10th day of the forecasting horizon when moving from the LSTM to the CNN. Furthermore, the CNN has shorter training and running times, providing such accurate results with a lower computational effort. The CNN provides similar results for the Lower Silesian and Opole Voivodeships. Its performances in the Greater Sidney Area are somehow less accurate while still good in absolute terms.

The same CNNs trained on PM$_{10}$ concentration can also provide reliable forecasts of PM$_{2.5}$ values, particularly in Lombardy and the Lower Silesian and Opole Voivodeships.

Finally, the CNN trained on the data of each area has also been used, without retraining, on the other two areas with satisfactory forecast accuracy.

This proves that the CNN structure proposed in this study has noticeable generalization capabilities and can thus be immediately used to forecast PM concentrations also in other contexts without the traditional long search for the best hyperparameter values. In other words, it can be considered a generic model, such as the process-based ones.

The proposed convolutional models show excellent forecasting performances, particularly for high values, representing the most critical health impact situations. However, some limitations of a simple convolutional architecture based on concentration measures remain and can probably be reduced by adding other inputs or modifying the network structure. For instance, one may add local meteorological variables or emission proxies such as population or traffic densities. On the other hand, one can test mixed convolutional–

LSTM structures to better capture some long-term trends or graph neural architectures that allow accounting for the semantic links between the stations' observations.

**Author Contributions:** Conceptualization, G.G.; methodology, L.F. and G.G.; formal analysis, L.F.; writing—original draft preparation, L.F. and G.G.; writing—review and editing, L.F. and G.G.; supervision, G.G.; validation, G.G. and L.F.; visualization, G.G. and L.F. All authors have read and agreed to the published version of the manuscript.

**Funding:** This research received no external funding.

**Institutional Review Board Statement:** Not applicable.

**Informed Consent Statement:** Not applicable.

**Data Availability Statement:** The data and codes used to perform this study are available at https://github.com/Lucaferra96/Convolutional_Forecasting_PM (accessed on 13 February 2024).

**Conflicts of Interest:** The authors declare no conflicts of interest.

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
