# Peer review of "Convolutional Forecasting of Particulate Matter: Toward a Data-Driven Generalized Model"

_atmosphere, doi:10.3390/atmos15040398_

Round 1

Reviewer 1 Report

Comments and Suggestions for Authors

Reviewer 2 Report

Comments and Suggestions for Authors

Dear Authors,

The article is interesting, however in order to improve the quality of the study it would be beneficial to make a correction before publication. I suggest considering the following comments:

1)      The abstract should emphasize what is new in the article. To what extent is the model developed by the authors better than others used in spatial analyses of PM concentrations in ambient air.

2) It is necessary to provide additional information on the adopted model. The issue of modeling and, in particular, forecasting pollutants in the ambient air for a period of 10 days is an extremely complex problem. The authors base their analysis on results from ground-based measurement stations. However, detailed information on the predictors adopted in the model is lacking. As is well known, the values of concentrations of particulate pollutants in the air are mainly influenced by: the volume of pollutant emissions depending on the emitters occurring in the analyzed areas, meteorological conditions (in particular, the wind along with its direction and speed, temperature - which is also related, for example, to emissions during the heating season, precipitation, terrain, etc.). The authors did not address these issues in the manuscript.

3) The results refer to average values for the entire areas analyzed? For a model averaged over the entire area, its utilitarian value is small. This is because the average value for the entire area may be relatively low, while locally concentrations may exceed limit values. This issue should be addressed.

4) Did the model validation (results presented) include the range of days for which the highest correlation between forecast and actual values was obtained? If so, what were the worst results for the periods with the lowest correspondence between model results and actual values?

5) The conclusions should address the limitations of the developed model regarding its ability to forecast PM2.5 and PM10 pollution.

6) More detailed information on the dominant sources of PM2.5 and PM10 emissions (and their shares) in the areas analyzed would be a beneficial addition to the article.

Minor comments:

- Chapters should not start with drawings (Figure 4).

- Avoid first-person descriptions in the manuscript, such as in line 340: "We show that...".

- It is necessary to unify in the content of the manuscript the accepted descriptions: PM2.5 and PM10 (in the description of the legend to Figures 1-3; Line 273); NRMSE (legend to Figures 5 and 7).

Best regards,

Round 2

Reviewer 1 Report

Comments and Suggestions for Authors

It can be accepted now 

Author Response

We thank the reviewer for the feedback.

Reviewer 2 Report

Comments and Suggestions for Authors

The authors of the manuscript have made significant corrections to it. However, my comments on specific errors were not fully addressed. Among other things, the format of the descriptions of the legends for Figures 1, 2 and 3 has not been standardized, and the designation "PM25" in the description of the legends for Figures 1 and 2 has not been corrected. In addition, the article contains minor editorial errors and, in my opinion, can be accepted for publication after additional proofreading.

Author Response

We are grateful for the useful reviewer’s feedback and comments. 

We fixed the legends of Figures 1, 2 and 3. Furthermore, we added further analysis and comments on the errors in the Lombardy domain in the Results section. Finally, we fixed some minor errors throughout the manuscript.